# Qutrit toric code and parafermions in trapped ions

Mohsin Iqbal[1,6], Anasuya Lyons[2,6], Chiu Fan Bowen Lo[2], Nathanan Tantivasadakarn [3], Joan Dreiling [4], Cameron Foltz[4], Thomas M. Gatterman [4], Dan Gresh[4], Nathan Hewitt [4], Craig A. Holliman[4], Jacob Johansen[4], Brian Neyenhuis[4], Yohei Matsuoka[4], Michael Mills [4], Steven A. Moses[4], Peter Siegfried [4], Ashvin Vishwanath [2], Ruben Verresen [2,5] & Henrik Dreyer [1] ✉

The development of programmable quantum devices can be measured by the complexity of many-body states that they are able to prepare. Among the most significant are topologically ordered states of matter, which enable robust quantum information storage and processing. While topological orders are more readily accessible with qudits, experimental realizations have thus far been limited to lattice models of qubits. Here, we prepare and measure a ground state of the $\mathbb{Z}_3$ toric code state on 24 qutrits (obtained by encoding one qutrit into two qubits) in a trapped ion quantum processor with fidelity per qutrit exceeding 96.5(3)%. We manipulate two types of defects which go beyond the conventional qubit toric code: a parafermion, and its bound state which is related to charge conjugation symmetry. We further demonstrate defect fusion and the transfer of entanglement between anyons and defects, which we use to control topological qutrits. Our work opens up the space of long-range entangled states with qudit degrees of freedom for use in quantum simulation and universal error-correcting codes.

The unprecedented tunability of quantum processors has opened up the on-demand preparation and control of topologically ordered (TO) quantum states[1–9]. Creating these states in programmable quantum computers does not only allow for the simulation of complex many-body systems, but also provides a code-space for quantum computation. In particular, the quasiparticle excitations of TO phases of matter —known as anyons—exhibit exchange statistics beyond those familiar from bosons or fermions[10–13]. The robust braiding of such anyons constitutes the primitive of topological quantum computation[14,15]. This experimental program is being extended to increasingly complex TOs, including non-Abelian TO[16–18] as well as defects that enrich the computational power of Abelian TO[19,20].

While anyons are pointlike deformations of the state, defects are associated to extended objects, like lattice dislocations or vortices in superfluids. As such, using defects to process quantum information in a topologically protected way is subject to more caveats than is the case of genuine anyonic excitations, as these rigid objects are harder to move and their braiding properties are more restrictive. On the other hand, defects can exist in comparatively less exotic states: A familiar example is the non-Abelian Majorana defect which can be inserted into the toric code[21], whereas Majorana anyons require a non-Abelian topologically ordered state[22]. In fact, defects can be seen as precursors to anyons. More precisely, defects are often related to global physical symmetries of the system and gauging these symmetries promotes defects into genuine anyons of a larger TO[23]. Thus, gauging introduces a hierarchy on the space of TOs[24].

For the conventional toric code[15], the only defects are the aforementioned Majoranas, associated with the $e \leftrightarrow m$ duality symmetry of

[1]Quantinuum, Leopoldstrasse 180, Munich, Germany. [2]Department of Physics, Harvard University, Cambridge, MA, USA. [3]Walter Burke Institute for Theoretical Physics and Department of Physics, California Institute of Technology, Pasadena, CA, USA. [4]Quantinuum, 303 S Technology Ct, Broomfield, CO, USA. [5]University of Chicago, Chicago, USA. [6]These authors contributed equally: Mohsin Iqbal, Anasuya Lyons. ✉e-mail: henrik.dreyer@quantinuum.com

the toric code, which exchanges the role of flux and charge and which is realized by mapping the square lattice to its dual or, equivalently, applying a Hadamard gate on each qubit and translating by half a lattice spacing. Here, and throughout this work, we will refer to the presence of anyons (like $e$ and $m$) whenever a stabilizer term in the Hamiltonian is violated, i.e., its expectation value is maximally different from the ground state value (+1). The hierarchy accessible through gauging is thus restricted. A richer class opens up by upgrading qubits to qutrits: a toric code based on the gauge group $\mathbb{Z}_3$ hosts two kinds of charges and fluxes; e.g., the electric $e$-anyon is no longer its own anti-particle, the latter now denoted as $\bar{e}$, and similarly for the flux anyons $m, \bar{m}$. Correspondingly, a new charge conjugation symmetry emerges: $e \leftrightarrow \bar{e}, m \leftrightarrow \bar{m}$. The resulting symmetry-enriched physics has received much attention, not least because it forms the backbone of certain proposals for obtaining universal non-Abelian error correction codes which can be prepared with constant-depth adaptive circuits[24–34]. Despite the theoretical interest, qudit-based TOs such as the $\mathbb{Z}_3$ toric code have so far eluded experimental observation both by devices with native qutrit degrees of freedom[35], as well as qubit-based platforms.

Here, we report the preparation of high-fidelity ground states of $\mathbb{Z}_3$ toric codes with periodic boundary conditions on up to $6 \times 4 = 24$ qutrits. We explicitly prepare two types of defects (Fig 1): a paraf-ermion defect (PF) and its conjugate (PF*)[36–40] which, similar to the $\mathbb{Z}_2$-case, are related to dislocations of the lattice. Further, we prepare a charge conjugation (CC) defect, which has no analog in the qubit toric code and is related to globally conjugating all charges and fluxes on-site. We verify their fusion rules by measuring their action on test anyons. Finally, we use these novel defects to produce a topological qutrit and initialize it by appropriately injecting long-range entanglement between two CC defect pairs. Experimentally, these advances are enabled by recent upgrades to Quantinuum's H2 ion-trap quantum computer allowing the use of 56 effectively all-to-all connected qubits, with two-qubit gate fidelities exceeding 99.8%[41,42]. These gate fidelities allow us to encode qutrit degrees of freedom into the native qubits of the device (cf. Supplementary Methods) and still achieve sufficiently high fidelities for the resulting one- and two-qutrit gates.

## Results

### Model & ground state preparation

To initialize our experiments, we first prepare the ground state of the rotated $\mathbb{Z}_3$ toric code. The Hilbert space consists of qutrit degrees of freedom, namely $|0\rangle, |1\rangle$, and $|2\rangle$, on the vertices of a square lattice with periodic boundary conditions. Similar to the simpler case of the $\mathbb{Z}_2$ toric code, we define stabilizers $A = \mathcal{X}^\dagger \mathcal{X} \mathcal{X}^\dagger \mathcal{X}$ and $B = \mathcal{Z} \mathcal{Z} \mathcal{Z}^\dagger \mathcal{Z}^\dagger$, which

act non-trivially on alternating plaquettes of the lattice (Fig. 2a)[15,43]. $\mathcal{X}$ and $\mathcal{Z}$ correspond to the qutrit clock matrices:

$$\mathcal{Z}|i\rangle = \omega^i |i\rangle \quad \text{and} \quad \mathcal{X}|i\rangle = |i+1\,(\mathrm{mod}\,3)\rangle \tag{1}$$

where $\omega = e^{2\pi i/3}$. While all $A$ and $B$ stabilizers commute due to the commutation relation $\mathcal{X}\mathcal{Z} = \omega \mathcal{Z}\mathcal{X}$, they are not Hermitian operators. To probe their expectation values, we consider the following projectors:

$$\Pi_A^\alpha = \frac{1}{3}(\mathbf{1} + \alpha^2 A + \alpha A^2) \quad \text{and}$$
$$\Pi_B^\alpha = \frac{1}{3}(\mathbf{1} + \alpha^2 B + \alpha B^2), \tag{2}$$

where $\alpha \in \{1, \omega, \bar{\omega}\}$.

The Hamiltonian for the $\mathbb{Z}_3$ toric code is defined as:

$$H = -\sum_{\substack{p \,\in\, \mathrm{type-A} \\ \mathrm{plaquettes}}} \Pi_{A_p}^1 - \sum_{\substack{p \,\in\, \mathrm{type-B} \\ \mathrm{plaquettes}}} \Pi_{B_p}^1. \tag{3}$$

The ground state subspace of the Hamiltonian is the simultaneous +1-eigenspace of the $A_p$ and $B_p$ stabilizers. In analogy to the familiar case of $\mathbb{Z}_2$ toric code, stabilizer violations on plaquettes indicate the presence of anyons. Specifically, a violation $\Pi_A^1 = 0$ (i.e., $A \neq +1$) on a type-A plaquette signals the presence of a charge anyon ($e$ or $\bar{e}$), while a violation $\Pi_B^1 = 0$ on a type-B plaquette indicates the presence of a flux anyon ($m$ or $\bar{m}$). The anyon type can be determined by measuring $\Pi_A^\omega$ and $\Pi_A^{\bar{\omega}}$: $(\Pi_A^\omega, \Pi_A^{\bar{\omega}}) = (1, 0)$ indicates the presence of an $e$ anyon, while $(\Pi_A^\omega, \Pi_A^{\bar{\omega}}) = (0, 1)$ signifies an $\bar{e}$ anyon. Similarly, flux anyons $m$ and $\bar{m}$ can be distinguished by measuring $\Pi_B^\omega$ and $\Pi_B^{\bar{\omega}}$. On a torus, the ground state subspace of $H$ is spanned by nine degenerate states. These states can be distinguished by the logical string operators $\mathcal{Z}_{\mathrm{hori}}$ and $\mathcal{Z}_{\mathrm{vert}}$, which are products of qutrit $\mathcal{Z}$ and $\mathcal{Z}^\dagger$ operators that wrap around the torus in the horizontal and vertical directions, and take values $1, \omega, \bar{\omega}$.

To prepare the logical $|00\rangle_L$ ground state characterized by $\mathcal{Z}_{\mathrm{hori}} = +1 = \mathcal{Z}_{\mathrm{vert}}$, we use the protocol described in ref.[44] and shown in Fig. 2b, c): The initial state of the $N$ qutrits in the quantum processor, $|0\rangle^{\otimes N}$, already fulfills $B = +1$ for all B-type plaquettes. We proceed by (i) choosing an A-type plaquette and a representative qutrit within it which is transformed into the state $|+\rangle := \frac{1}{\sqrt{3}}(|0\rangle + |1\rangle + |2\rangle)$, and (ii) applying a sequence of controlled-$\mathcal{X}$ (C$\mathcal{X}$) or C$\mathcal{X}^\dagger$ gates to the remaining qutrits within the plaquette, with the choice of gate (C$\mathcal{X}$ or C$\mathcal{X}^\dagger$) determined by whether the target qutrit is acted upon by $\mathcal{X}$ or $\mathcal{X}^\dagger$ in the stabilizer $A$ (Fig. 2c). The action of the C$\mathcal{X}$ gate on two qutrits is

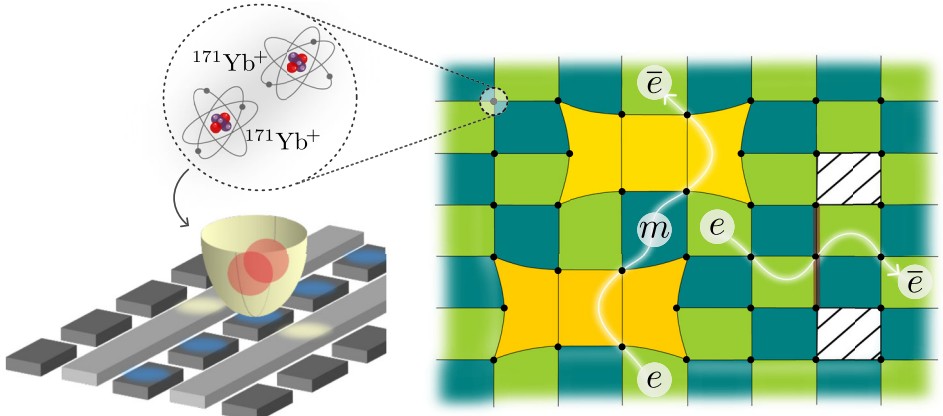

**Fig. 1 | Concept.** We start with 56 trapped $^{171}$Yb$^+$ ions in a quantum charge-coupled device and algorithmically encode pairs of qubits into qutrits. This allows us to prepare ground states of $\mathbb{Z}_3$ toric codes on tori of up to $6 \times 4$ qutrits. We conduct experiments to study the relationship between the anyons and the topological defects of this system, namely parafermion (shaded yellow) and charge conjugation defects (hatch pattern).

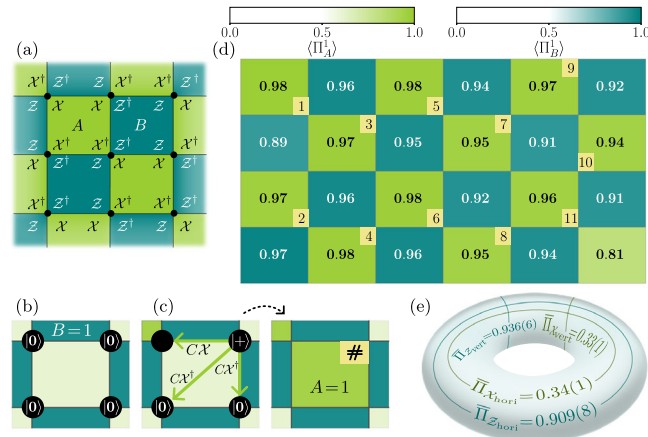

**Fig. 2 | Preparation of qutrit toric code. a** Square lattice on a torus with qutrits on the vertices. **b** Qutrits are initialized in the $|0\rangle$ state, satisfying $B = +1$ (visually represented by the intense turquoise color of type-B plaquettes, in contrast to the faded green color of type-A plaquettes, which do not satisfy the $A = +1$ condition at this stage). **c** Preparation of one of the type-A plaquettes. One of the qutrits is initialized in the $|+\rangle$ state. This qutrit is used as a control, and we apply $C\mathcal{X}$ or $C\mathcal{X}^\dagger$ gates to other qutrits in the plaquette. This leads to satisfying $A = +1$, indicated by a bright green color on the right-hand side of the arrow. A small square in one corner of the plaquette indicates the control qutrit. The number within the square denotes the order in which the corresponding stabilizer is prepared. **d** Expectation values of projectors $\Pi_\bullet^1$ obtained by measuring qutrits in the $\mathcal{X}$ and $\mathcal{Z}$ basis. The maximum error in estimating the expectation values is 0.022. The mean energy density, $\langle H \rangle / 24 \geq -1$, is found to be $-0.945(3)$. **e** Mean expectation values of projectors for the logical $\mathcal{X}$ and $\mathcal{Z}$ operators in two directions on the torus closely match the theoretical predictions: $\langle \Pi_{\mathcal{Z}_{\text{hori}}}^1 \rangle = \langle \Pi_{\mathcal{Z}_{\text{vert}}}^1 \rangle = 1$ and $\langle \Pi_{\mathcal{X}_{\text{hori}}}^1 \rangle = \langle \Pi_{\mathcal{X}_{\text{vert}}}^1 \rangle = \frac{1}{3}$. All error bars in this work denote one standard error on the mean.

$C\mathcal{X}|i,j\rangle = |i, i+j\,(\mathrm{mod}\,3)\rangle$. We repeat steps (i) and (ii) until all but one A-type plaquettes have been chosen, while carefully avoiding to designate a qutrit as representative that has previously been acted on by a $C\mathcal{X}$ gate (see Fig. 2d for our chosen ordering). The final plaquette is implicitly prepared due to the symmetry constraint on the operators $\prod_p A_p = 1$.

At the end of the circuit, we measure all qutrits in both the $\mathcal{X}$ and $\mathcal{Z}$ bases to compute the expectation value of $\Pi_\bullet^1$ for every plaquette. A barrier is inserted before performing destructive qutrit measurements which ensures that the entire quantum state is prepared before the measurements collapse the wavefunction into single qutrit eigenstates. As our qutrit encoding uses two qubits per qutrit, the remaining one-dimensional subspace can be used to detect errors during preparation that cause qutrits to leak outside of the qutrit subspace. These errors are heralded, and the corresponding shots are discarded, representing ~11% of the total number of shots. The values presented in the main text are computed from the remaining, retained shots (cf. Supplementary Note 2 for the raw data for different system sizes).

To assess the quality of the prepared state, we show the expectation value of $\Pi_\bullet^1$ for each plaquette, as well as the logical operators in Fig. 2d, e. The logical mean values were computed by averaging across columns for horizontal operators and across rows for vertical operators. Measurement of the correlations between stabilizers of a given type allow us to bound the fidelity per site with the logical $|00\rangle_L$ state, detailed in Supplementary Note 1, as

$$0.965(3) \leq \left(\langle 00|_L \rho |00\rangle_L\right)^{1/24} \leq 0.984(2) \qquad (4)$$

and the lower bound further increases to 0.974(3) after accounting for readout errors, as discussed in Supplementary Note 2.

We observe that the expectation value of $\Pi_{A_p}^1$ for the implicitly prepared plaquette (i.e., the bottom right plaquette in Fig. 2d) is

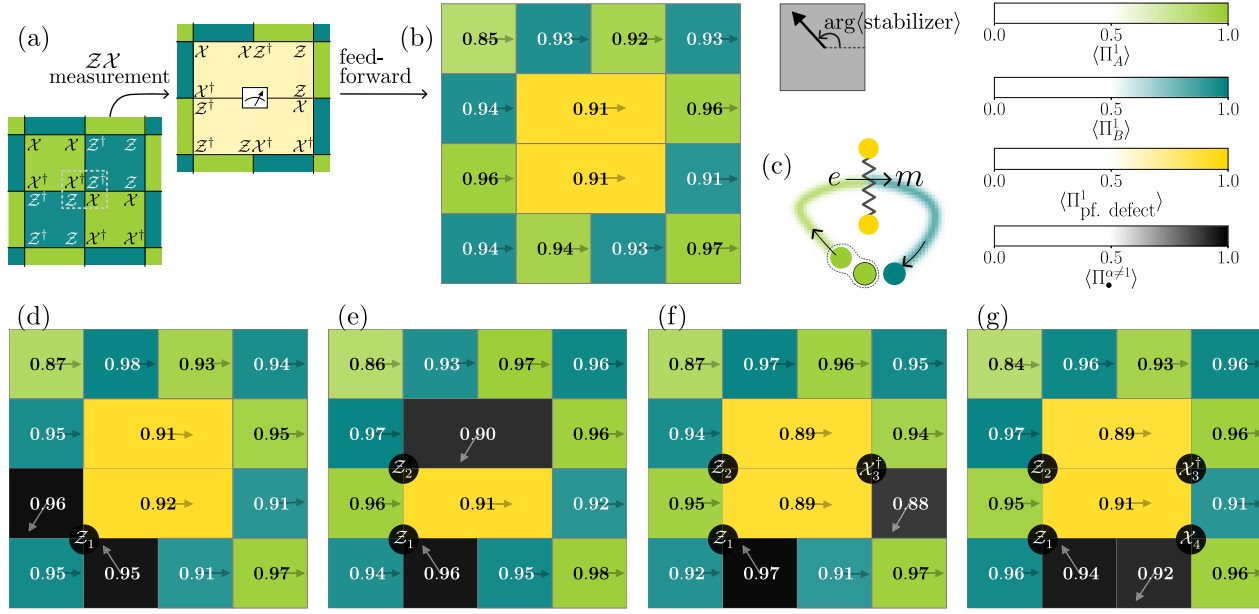

**Fig. 3 | Creation of and braiding around parafermion defects.** Any plaquette containing an anyon is colored black, with the value of $\max(\Pi_\bullet^\omega, \Pi_\bullet^\omega)$ displayed. An arrow within each plaquette indicates the direction specified by the arg $\langle$stabilizer$\rangle$, where the stabilizer could be $A_p$, $B_p$, or any of the defect stabilizers. The arrow's direction serves as a visual cue to distinguish anyons from their conjugates. **a, b** A pair of defects is inserted into the ground state by measuring the middle qutrit in the $\mathcal{X}\mathcal{Z}$-basis and performing feed-forward based on the measurement outcome.

**c** A sketch illustrating the braiding experiment in steps (**d–g**). A pair of charges, $e$ and $\bar{e}$, is created by applying a $\mathcal{Z}$ operator, which toggles the eigenvalues of the neighboring green ($\mathcal{X}$)-type plaquettes. Charge $\bar{e}$ remains fixed, while $e$ is dragged through the defect pair and emerges as $m$ on the other side of the defect pair, signaled by the fact a blue ($\mathcal{Z}$-type) plaquette is now excited. The maximum estimation error is 0.022.

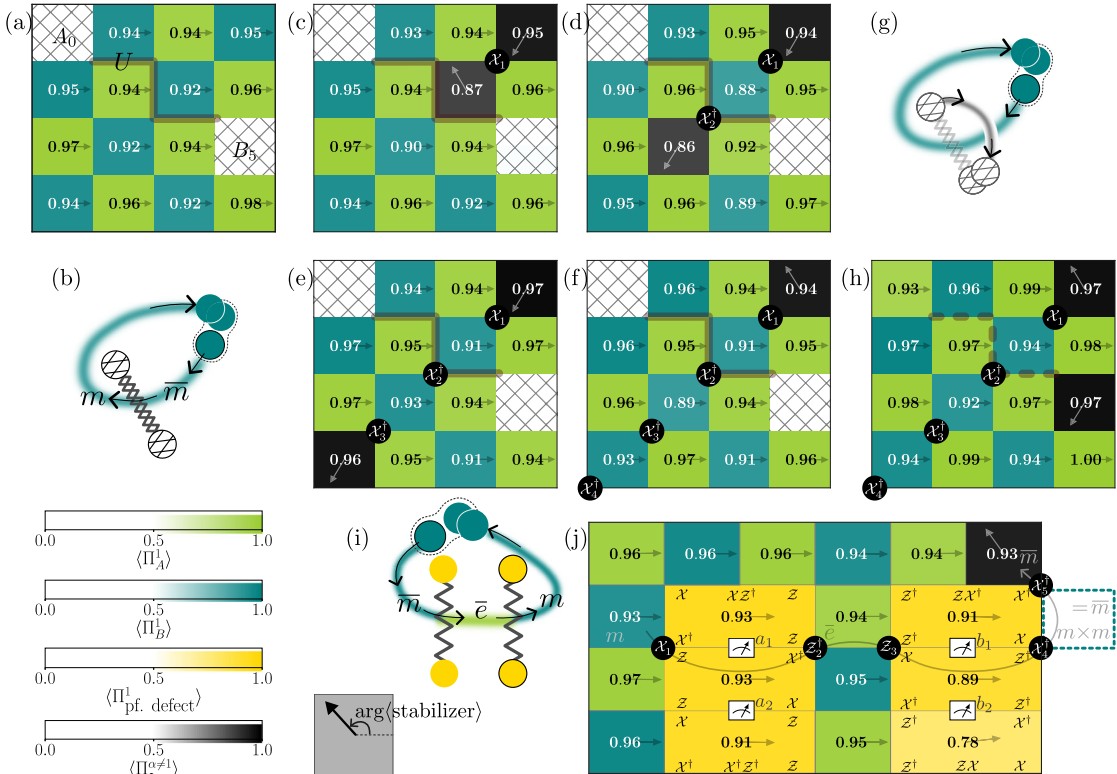

**Fig. 4 | Creation and braiding of CC defects and their relation to parafermions.** **a** Ground state of the $\mathbb{Z}_3$ toric code with a CC defect pair. The endpoints of the thick line, representing the CC construction circuit $U$, correspond to high-weight stabilizers $A_0$ and $B_5$ (defined in Supplementary Fig. 9c). These stabilizers are marked by a hatching pattern with a ' × ' symbol, and their values are omitted for clarity as they label the internal state of the defects which is not locally accessible. **b** A sketch of the braiding experiment in (**c**–**f**). A flux pair $m - \bar{m}$ is created. $\bar{m}$ is transmuted into $m$ by commuting it through the CC defect line, and it is then fused with the fixed $m$ anyon at the top right corner through a sequence of four steps (**c**–**f**), by applying

$\mathcal{X}_1, \mathcal{X}_2^{\dagger}, \mathcal{X}_3^{\dagger}$, and $\mathcal{X}_4^{\dagger}$. **g** Outline of the braiding experiment in (**h**) where the CC defect pair is fused. This is achieved by applying the same circuit $U$ used in (**a**) (see Supplementary Note 5). The dashed line shows the path (as implemented by $U$) taken to fuse the defect pair. The altered state of the CC defect pair is revealed as a flux $m$ at one endpoint. **i** A sketch of the braiding experiment in (**j**). We prepare the ground state and create two parafermion defect pairs. The $m$ flux from the pair $m - \bar{m}$ created in the second plaquette from the top, leftmost corner remains fixed, while its partner $\bar{m}$ anyon is commuted through two parafermion defect pairs. The resulting $m$ is then fused with the pinned $m$ to give a single $\bar{m}$ anyon.

generally slightly lower than the expectation values of the remaining type-A plaquettes, which are explicitly prepared. This observation can be attributed to the fact that, if a charge pair ($e - \bar{e}$) is created due to a $\mathcal{Z}$ or $\mathcal{Z}^{\dagger}$ error during state preparation, the sequential preparation of plaquettes will drag one of these spurious anyons all the way to final plaquette. This is in contrast to non-unitary preparation schemes where no significant translation symmetry breaking has been observed even when subjected to noise[4,5].

**Parafermion defects**

Equipped with a high-fidelity ground state, we turn to the study of topological defects. One type of defect supported by the $\mathbb{Z}_3$ toric code generalizes the well known *em*-defect of its $\mathbb{Z}_2$ counterpart. That defect acts on the flux and charge anyons by exchanging their identities and behaves like a Majorana[22,45], which has been used experimentally to implement logical Clifford gates on a $\mathbb{Z}_2$ toric code background[19,20]. In contrast, for the $\mathbb{Z}_3$ toric code, a generalization of such a defect has been predicted to have parafermion fusion rules[46]

$$\text{PF} \times \text{PF} = \mathbf{1} + \bar{e}m + e\bar{m}. \tag{5}$$

By fusion outcome, we mean the resultant stabilizer measurement when two excitations are moved to the same location on the lattice. As we will see, two distinct species of parafermions, labeled PF and PF* can arise in the $\mathbb{Z}_3$ case.

To deterministically prepare a pair of PF defects we first initialize the ground state and then measure one of the qutrits in the basis in which either the operator $\mathcal{X}\mathcal{Z}$ or $\mathcal{X}\mathcal{Z}^{\dagger}$ (corresponding to the distinct parafermion species) is diagonal. We apply a feedforward operation based on the measurement outcome which ensures that the corresponding defect stabilizers have definite values. Specifically, when the measurement outcome is 1, we apply a $\mathcal{Z} \otimes \mathcal{Z} \otimes \mathcal{Z}^{\dagger}$ to the left, bottom-left and bottom-center qutrit in Fig. 3a. If the measurement outcome is 2, the conjugate is applied instead and no action in case of a 0 measurement outcome. Having created the ground state with an initialized em-defect pair (cf. Figure 3b), we create a charge-anticharge ($e - \bar{e}$) pair and subsequently braid $e$ around one of the defects using the qutrit clock matrices (Fig. 3d–g). Upon passing through the line connecting the defects, a change in stabilizer expectation values indicates that a permutation

$$e \rightarrow m \tag{6}$$

has occurred. Finally, at the end of the experiment, we are left with a single dyon $\bar{e}m$, i.e., the fusion outcome in (5) has been toggled from the identity to the dyon channel. This demonstrates the parafermion behavior predicted for the $\mathbb{Z}_3$ toric code.

**Charge conjugation defects and relation to parafermions**

The richer anyon content of the $\mathbb{Z}_3$ toric code allows for another type of topological defect, which has no analog in the $\mathbb{Z}_2$ case: As an anyon

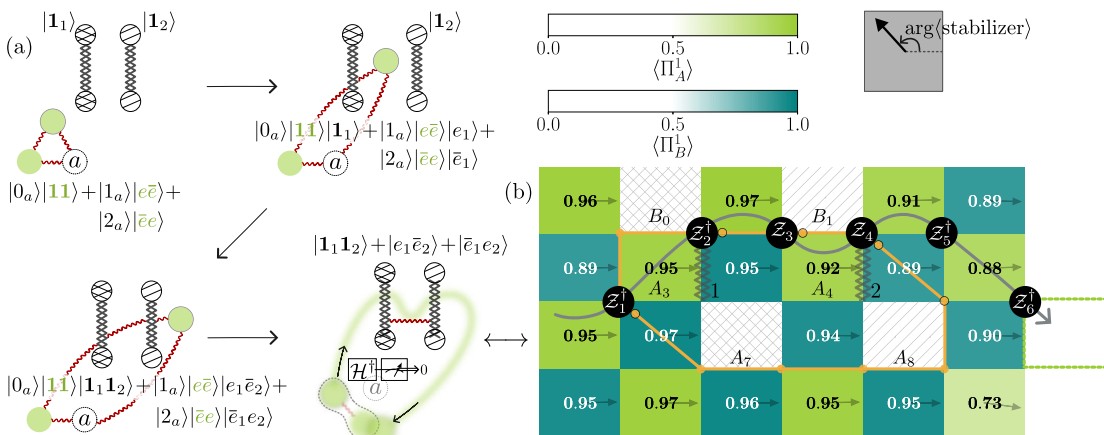

**Fig. 5 | Entanglement transfer from anyons to CC defects for initializing topological qutrits. a** A sketch of the different steps, with intermediate states, involved in moving a charge anyon around defects. The braiding followed by measuring an ancilla transfers a Bell state of charge anyons into an entangled logical state of CC defects. **b** Results for the final step as depicted in (**a**). We create $\mathbb{Z}_3$ ground state with two pairs of defects, labeled 1 and 2 (cf. Supplementary Fig. 10). Defect pair 1, marked with the ×-hatch pattern, extends between points $B_0$ and $A_7$. Defect pair 2, indicated by the /-hatch pattern, has endpoints $B_1$ and $A_8$. The orange loop represents a braid that stabilizes the prepared topological qutrit state; a black border on the solid circle indicates the application of $\mathcal{X}^\dagger$, while its absence indicates $\mathcal{X}$. The expectation values of the projectors $\Pi^1_{B_0}$ and $\Pi^1_{B_1}$ for the non-local stabilizers are 0.931(15) and 0.927(15), respectively. We measure the expectation values of $\Pi^1_{A_7}$, $\Pi^1_{A_8}$, and $\Pi^1_{A_7 A_8}$ for different ancilla outcomes ($|0_a\rangle$, $|1_a\rangle$, and $|2_a\rangle$). For $\Pi^1_{A_7}$, the measured values are 0.44(5), 0.40(5), and 0.39(5), respectively. Similarly, for $\Pi^1_{A_8}$, the values are 0.44(5), 0.42(5), and 0.36(5). Finally, $\Pi^1_{A_7 A_8}$ yields values of 0.81(4), 0.78(4), and 0.74(4) for the respective ancilla states. This is a manifestation of the fact that, although the outcomes for each individual defect pair are random, they are jointly in an entangled state.

traverses a line connecting a pair of such charge conjugation (CC) defects, it is turned into its antiparticle, i.e., $e \leftrightarrow \bar{e}$ and $m \leftrightarrow \bar{m}$. Experimentally demonstrating the action of this novel type of defect is what we turn to next.

To this end, we apply a circuit $U$ (cf. Supplementary Fig. 9b) to the ground state of the $\mathbb{Z}_3$ toric code. A key ingredient of the CC defect pair unitary construction is the one-qutrit charge conjugation gate, which acts as $\mathcal{C}|i\rangle = |-i\,(\mathrm{mod}\,3)\rangle$. Intuitively, the circuit construction unzips the $\mathbb{Z}_3$ toric code to a trivial paramagnet, applies $\mathcal{C}$ to this trivial state, and then returns to the $\mathbb{Z}_3$ toric code—see Supplementary Note 4 as well as[47] for a more in-depth derivation. We then create a pair of $m - \bar{m}$ anyons and move $\bar{m}$ through the line connecting the CC defect pair. This operation transforms

$$\bar{m} \to m, \tag{7}$$

as evidenced by the change in the direction given by arg$\langle$stabilizer$\rangle$, which is visually represented by an arrow within each plaquette in Fig. 4. The arrow direction changes approximately from 120° to 240° in the excited plaquettes on two sides of the CC defect line (Fig. 4(c, d)). The transformed anyon ($m$) is then transported around the torus and fused with its partner (another $m$), resulting in a single $\bar{m}$ particle (Fig. 4(e, f)).

Crucially, as the $\bar{m}$ particle traverses the defect line, it alters the internal state of the CC defect pair which is invisible for local observables. This altered state manifests itself as an $m$ anyon (cf. Figure 4h) upon coherently moving and fusing back the CC defect pair by applying the same unitary $U$ used in Supplementary Fig. 9b (cf. Supplementary note 5 for why applying $U$ again achieves coherent movement of the end of a CC defect), demonstrating an instance of the fusion rules

$$\mathrm{CC} \times \mathrm{CC} = (\mathbf{1} + \bar{e}m + e\bar{m})(\mathbf{1} + em + \bar{e}\bar{m}). \tag{8}$$

We will later exploit the ability of the CC defect pair to store information in its fusion channel, which allows for the distribution of entanglement in a non-local way.

Having demonstrated the action of parafermion (6) and CC defects (7) on the anyons, we are now in a position to demonstrate their mutual fusion rule, namely,

$$\mathrm{PF} \times \mathrm{PF}^* = \mathrm{CC}, \tag{9}$$

i.e., the combined action of a parafermion defect and its conjugate is equivalent to that of a charge conjugation. Note that (9) is compatible with multiplying (5) with its conjugate and comparing to (8). To illustrate this, we generate two pairs of parafermion defects (cf. Figure 4j). For the first pair, we measure the qutrit $a_1$ in the $\mathcal{X}\mathcal{Z}$ basis and the qutrit $a_2$ in the $\mathcal{X}\mathcal{Z}^\dagger$ basis, resulting in a pair of PF defects. Similarly, for the second pair, we measure qutrit $b_1$ in the $\mathcal{X}\mathcal{Z}$ basis and qutrit $b_2$ in the $\mathcal{X}\mathcal{Z}^\dagger$ basis, producing a pair of PF* defects. Qutrit measurements are followed by a real-time feedforward operation to deterministically initialize the parafermion defects. Next, we act with $\mathcal{X}_1$, which creates an $m$ anyon in a type-A plaquette and simultaneously excites the inner defect plaquette of the left parafermion defect pair (Fig. 4j). We then move one of the anyons to the right, wrapping around the torus. The left parafermion defect pair transmutes $\bar{m}$ to $\bar{e}$, while the right parafermion defect pair changes $\bar{e}$ to $m$. This $m$ anyon is then fused with its partner, the original $m$ anyon, leaving us with a single $\bar{m}$ anyon. This is the exact same outcome we obtained for a similar anyon trajectory and a single pair of CC defects, which would similarly transform an $\bar{m} \to m$ when moved across. In addition, Supplementary Fig. 8 demonstrates the conjugation of (part of) a parafermion defect line with a CC defect line, effectively resulting in a conjugate parafermion defect line.

## A topological qutrit

Finally, having established control over the anyons and defects of the $\mathbb{Z}_3$ toric code, we use these ingredients to produce a topological qutrit. ref.48 gives the standard definition of a topological qutrit as a collection of four non-Abelian quasiparticles whose overall fusion outcome is neutral but individual pairs have three possible fusion outcomes.

The non-Abelian objects we use to create such a qutrit are given by two pairs of charge conjugation defects, created from the toric code vacuum. In principle, a pair of defects can have nine different fusion

outcomes, according to Eq. (8), representing two qutrits worth of information. For simplicity, here we will focus on the single-qutrit subspace spanned by

$$|\phi_1\rangle = \left[|\mathbf{1}_1\mathbf{1}_2\rangle + |e_1\bar{e}_2\rangle + |\bar{e}_1 e_2\rangle\right]/\sqrt{3}$$
$$|\phi_\omega\rangle = \left[|\mathbf{1}_1\mathbf{1}_2\rangle + \omega|e_1\bar{e}_2\rangle + \bar{\omega}|\bar{e}_1 e_2\rangle\right]/\sqrt{3} \qquad (10)$$
$$|\phi_{\bar{\omega}}\rangle = \left[|\mathbf{1}_1\mathbf{1}_2\rangle + \bar{\omega}|e_1\bar{e}_2\rangle + \omega|\bar{e}_1 e_2\rangle\right]/\sqrt{3}$$

where $|\alpha_1\beta_2\rangle$ denotes the state in which the left (right) defect pair fuses to $\alpha$ ($\beta$). That is, we restrict to the magnetically neutral sector in which no $m$ or $\bar{m}$ anyons appear in any of the intermediate fusion outcomes.

The logical $Z_L$ and $X_L$ operators on this topological qutrit can be realized on the physical level as follows: Creating a charge-anti-charge pair and moving the charge through both defect lines (e.g., using the path given by the gray line in Fig. 5b) implements a logical $Z_L = |\mathbf{1}_1\mathbf{1}_2\rangle\langle e_1\bar{e}_2| + |e_1\bar{e}_2\rangle\langle\bar{e}_1 e_2| + |\bar{e}_1 e_2\rangle\langle\mathbf{1}_1\mathbf{1}_2|$. Similarly, braiding a flux around one of the defect pairs picks up a phase depending on the internal state of that defect and thus this operators realizes a logical $X_L = |\phi_1\rangle\langle\phi_\omega| + |\phi_\omega\rangle\langle\phi_{\bar{\omega}}| + |\phi_{\bar{\omega}}\rangle\langle\phi_1|$.

However, $X_L$ or $Z_L$ cannot initialize the topological qutrit states (10) starting from $|\mathbf{1}_1\mathbf{1}_2\rangle$. To do this, we demonstrate the application of a logical Fourier transform $\mathcal{H}$ (cf. Supplementary Methods): A superposition of states with physical anyons pairs $e\bar{e}$, and $\bar{e}e$ injected at a fixed position and the vacuum is produced using a control-$\mathcal{Z}$ operation conditioned on an ancilla qutrit '$a$' initialized in the state $(|0\rangle_a + |1\rangle_a + |2\rangle_a)/\sqrt{3}$. One of the anyons is then coherently moved to braid around one half of each defect pair before being annihilated with its partner. The intermediate states of the ancilla, charge anyons, and defect pairs are depicted in Fig. 5a. After applying an additional $\mathcal{H}^\dagger$ gate (cf. Supplementary Methods) on the ancilla, the resulting state of defect pairs and ancilla is proportional to:

$$|0\rangle_a|\phi_1\rangle + |1\rangle_a|\phi_\omega\rangle + |2\rangle_a|\phi_{\bar{\omega}}\rangle. \qquad (11)$$

After measuring the ancilla and recording the measurement outcome, we have prepared the logical state $|\phi_{\omega^j}\rangle$ where $j$ is the measurement outcome of the ancilla. A logical $X_L$ operation can be used to complete the state preparation protocol if deterministic state preparation is desired.

Crucially, the entanglement between the system and the ancilla has been transferred from a local to a non-local information carrier and is now robust: As long as the distance between both the endpoints of each of the defects as well as between the defect lines themselves is sufficiently large, any anyons created by a local noise process can maximally encircle a single endpoint. However, any process in which there is an odd number of anyons crossings the defect lines will result in an odd number of charge or flux anyons. By fusing the spurious anyon back into the closest defect, we can return to the original logical state.

To certify the non-local entanglement of the defect pairs, we focus on the shots where the ancilla has been measured in the $|0\rangle$ state and make use of the fact that $|\phi_1\rangle$ is uniquely specified by being a +1 eigenstate of two commuting anyon braids: The first braids a flux around *both* defect pairs (denoted by the orange line in Fig. 5b) and is microscopically implemented by a string of physical $\mathcal{X}$ and $\mathcal{X}^\dagger$. The second anyon braid is simply the logical $Z_L$-operator defined above. Measuring the expectation values of these operators leads to fidelity bounds

$$0.72(5) \leq \text{Tr}[\langle\phi_1|\rho|\phi_1\rangle] \leq 0.80(4), \qquad (12)$$

and we report the results for the different states $|\phi_{\omega^j}\rangle$ as well as SPAM-corrected values in Supplementary Note 1. All states are prepared with

fidelities that far exceed those that can be reached with a classical mixture, which certifies that the two fusion channels of the defect pairs have been successfully entangled. For comparative analysis, Supplementary Note 3 presents results for a $6 \times 2$ lattice geometry, providing a more detailed description of the entire procedure. We note that other logical states, like $|\phi_0\rangle + |\phi_\omega\rangle + |\phi_{\bar{\omega}}\rangle \propto |\mathbf{1}_1\mathbf{1}_2\rangle$ are easier to achieve than what we have presented since they do not require the ancilla-assisted logical Fourier transform gate we have executed.

## Discussion

We have created high-fidelity ground states of the $\mathbb{Z}_3$ toric code on tori of sizes up to $6 \times 4 = 24$ qutrits using Quantinuum's H2 trapped ion quantum computer. We found that the unitary state preparation scheme employed in this work shows a much higher degree of translation-symmetry breaking than measurement-based preparation schemes, which could be used in heralded encoding protocols in future work. We have created parafermion and charge conjugation defect pairs on top of this vacuum and verified their action on the anyons of the model; some of these operations were facilitated by using adaptive quantum circuits. Finally, we have initialized a topological qutrit from two charge conjugation defect pairs. An appealing direction for future work is to explore syndrome measurements and repeat-until-success protocols, which have thus far led to a break-even for state preparation and measurement errors for simpler code states[6,7,49–51].

Our findings show that digital quantum processors have advanced to the point where they can throw off the shackles of the underlying qubit architecture and explore the much larger class of systems that is naturally formulated with local Hilbert space dimensions greater than two, which includes lattice gauge theories[52–55] and models with spin greater than one-half[56,57]. While our results show that qutrit quantum simulations are now generally possible using qubit-based devices, the question of whether fault-tolerance thresholds for qutrit-based codes can be achieved in such devices remains an open question. Tantalizingly, that class includes models with universal quantum computational power, some of which are closely related to the model presented here[25].

## Data availability

The data generated in this study have been deposited in the Zenodo repository https://doi.org/10.5281/zenodo.14007593[58]. database under open access.

## Code availability

The code used for numerical simulations is available from from Zenodo repository https://doi.org/10.5281/zenodo.14007593[58].

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

## Acknowledgements

We thank the broader team at Quantinuum for comments. N.T. is supported by the Walter Burke Institute for Theoretical Physics at Caltech. A.V. and R.V. are supported by the Simons Collaboration on Ultra-Quantum Matter, which is a grant from the Simons Foundation (618615, A.V.). A.L. and C.F.B.L. acknowledge support from the National Science Foundation Graduate Research Fellowship Program (NSF GRFP). This work is in part supported by the DARPA MeasQuIT program. The experimental data in this work were produced by the Quantinuum H2 trapped ion quantum computer, powered by Honeywell, in 2024.

## Author contributions

J.D., C.F., T.M.G., D.G., N.H., C.A.H., J.J., B.N., Y.M., M.M., S.A.M. and P.S. ran the experiment and took the data. M.I., A.L., C.F.B.L., N.T., A.V., R.V. and H.D. conceived the experiments, translated the ideas to quantum circuits, did the data analysis and drafted the manuscript.

## Competing interests

H.D. is a shareholder of Quantinuum. All other authors declare no competing interests.
