## [Transparent Peer Review file · Nature Communications]

Qutrit Toric Code and Parafermions in Trapped Ions

Corresponding Author: Dr Henrik Dreyer

Version 0:

Reviewer comments:

Reviewer #1

(Remarks to the Author)
Qutrit Toric Code Review:

Key Results

The authors provide a novel and timely study of a qudit based algorithm for quantum computation of the Z_3 toric code using and industry platforms trapped ion quantum computer. The authors prepare a high fidelity topological state across 24 qutrits and demonstrate that certain operations manipulating topological effects correspond to qutrit operations.

Validity

I can find no faults in the data analysis that the authors have done. The results of their work appear sound.

Significance

The results are quite novel and one of the more compelling and interesting quantum simulations I have seen in the last year. The benefits of this work will likely extend to development of fault-tolerant qudit simulations which will be of great use to a wide variety of fields. If the work continues along side algorithmic developments for qudit based hardware I suspect there will be continued promise for utility scale qudit-based quantum computers. I recommend publication with minimal changes

Analytical Approach

The study of the plaquette model and ground state preparation algorithm is technically correct as written. The figures in 2a and 3a accurately demonstrate the qutrit toric code and are an excellent supplement to the existing text. The averaging procedures the authors use to identify the effective accuracy of the prepared states are sound, however certain pieces of the analysis are missing for complete trust in the accuracy of the error bounds quoted such as in Eq. (4) and the abstract. Currently a detailed explanation on how significant correlated readout errors are is missing however I do not expect this to substantially change the final results.

Suggested improvements

I cannot find many improvements from a scientific perspective for this work. I enumerate my minor concerns below.

On page 2 "A barrier is inserted before performing destructive qutrit measurements which ensures the entire quantum state is prepared..." Can the authors indicate more clearly that the barrier is a compiler command (if this is a correct statement).

On page 3: I really like figure 2. I would suggest adjusting the second line on the torus image because the blue is slightly hard to read against the dark grey.

On page 4: I would like a larger and broader discussion on the impacts for quantum computing in terms of systems that would benefit from qutrit based hardware, i.e. high energy physics, spin models etc.

Reviewer #2

(Remarks to the Author)

The paper by Iqbal et al presents an emulation of a qutrit toric code based on using pairs of qubits to encode each qutrit. Using up to 56 qubits, they study the topological properties of the toric codes as well as defects within the code. The manuscript demonstrates interesting physics and very nice experimental results in a regime that has so far been challenging to reach for quantum computers. This highlights the potential of near-term devices for the study of exotic phases of matter. However, the presentation is at times quite difficult to follow, since topics such as anyon braiding are very specialized. For a broad journal such as Nature Communications, a more pedagogical introduction is needed to fully grasp the results. More detailed comments below.

1) The title and abstract should be transparent about the fact that this is an emulation of a qutrit experiment, rather than a native qutrit experiment. With the recent push into qudit computing platforms, I believe this distinction is relevant so that the article will not be misunderstood. The actual hardware implementation is first mentioned only at the end of Sec 1, which is too late.

2) Line 59: "e \leftrightarrow m symmetry of the toric code" is undefined. What does e and m refer to here? Also, more explanation is needed regarding the anyon properties and operations. e.g. how is Eq. (5) and (8) to be understood?

3) The color shading in the figures is chosen over a range from 0 to 1, while the values are generally between 0.9 and 1. Hence, the differences in shading are almost undetectable. I suggest shifting the range to make this more useful.

4) How are 56 ions enough to emulate 24 qutrits, which need 2 ions each? Could be a typo, or maybe I am missing something?

5) Sec III / Fig. 3: More explanation is needed for the feed forward operation used in Fig.3a. More explanation is needed for the braiding experiment in Fig.3. More explanation is needed on how the application of a Z operator creates a charge anticharge pair. The change e \rightarrow m mentioned in Eq.(6) appears to be simply due to the location of the charge: on green plaquettes it is e, on blue plaquettes it is m? Under what circumstances should one expect a different outcome here?

6) Sec IV: More explanation needed for the hatching in Fig. 1a. Why are these values not defined after the circuit? More explanation is needed for Fig.4h. In Fig 4j, it is mentioned that the pair is created at the top left corner, yet in the figure it appears to be one plaquette lower, which leads to confusion with the final fusion of the pair. In line 284 it is stated that the same outcome is obtained using "a single pair of CC defects." Please clarify what is referred to here. In the next line it is stated that Fig.13 "partially demonstrates the conjugation of ..." - why partially?

7) Sec V: The state description in line 305 is unclear. What does it mean that a "pair fuses to" e?

8) It would be nice to add some discussion about what one learns from these experiments.

Version 1:

Reviewer comments:

Reviewer #1

(Remarks to the Author)

I stand by my prior review and believe this work is worth publication in nature communications.

Reviewer #2

(Remarks to the Author)

The revised manuscript has improved in several aspects in terms of clarity. This should help non-experts to follow the physics better. Overall, the results are a nice demonstration of the potential of today's quantum computing devices and should be interesting to a broad audience.

Reviewer #1:

Qutrit Toric Code Review:

Key Results

The authors provide a novel and timely study of a qudit based algorithm for quantum computation of the Z3 toric code using and industry platforms trapped ion quantum computer. The authors prepare a high fidelity topological state across 24 qutrits and demonstrate that certain operations manipulating topological effects correspond to qutrit operations.

Validity

I can find no faults in the data analysis that the authors have done. The results of their work appear sound.

Significance

The results are quite novel and one of the more compelling and interesting quantum simulations I have seen in the last year. The benefits of this work will likely extend to development of fault-tolerant qudit simulations which will be of great use to a wide variety of fields. If the work continues along side algorithmic developments for qudit based hardware I suspect there will be continued promise for utility scale qudit-based quantum computers. I recommend publication with minimal changes

Analytical Approach

The study of the plaquette model and ground state preparation algorithm is technically correct as written. The figures in 2a and 3a accurately demonstrate the qutrit toric code and are an excellent supplement to the existing text. The averaging procedures the authors use to identify the effective accuracy of the prepared states are sound, however certain pieces of the analysis are missing for complete trust in the accuracy of the error bounds quoted such as in Eq. (4) and the abstract. Currently a detailed explanation on how significant correlated readout errors are is missing however I do not expect this to substantially change the final results.

We thank the reviewer for their detailed and positive feedback. Regarding the analysis of the error bound in Equation (4), these bounds are rigorous for the overall fidelity of the whole state preparation and measurement sequence (as derived in “Appendix B: Fidelity Bounds”). Now, distinguishing between State Preparation and Measurement errors in quantum setups is fundamentally challenging (which is why usually the composite SPAM metric is reported).

One thing one can do is to disentangle the errors in the state preparation sequence from errors that would occur even if no gates are applied at all. This is the SPAM mitigation analysis that we have done in “Appendix C: Ground State Preparation Data Analysis”. Note however, that the quoted number of 96.5(3)% in the abstract is the lower bound *without* any attempt at SPAM error mitigation and is thus the fidelity taking into account **all** errors including measurement, for

which Eq. (4) is a rigorous bound.

To indicate that the rigorous bounds of Eq. (4) refers to such a composite SPAM metric, we have modified the abstract: *“Here, we prepare and measure a ground state of the Z_3 toric code state on 24 qutrits (obtained by encoding one qutrit into two qubits) in a trapped ion quantum processor with fidelity per qutrit exceeding 96.5(3)%”.*

Regarding correlated readout errors: Qubit measurements are performed by state-selective fluorescence detection, with each qubit measured in remote locations (with separation much larger than the wavelength of the measurement light). The most plausible mechanism by which a measurement error on qubit A (or the outcome of its measurement at all) could impact the measurement outcome of another qubit B is the rescattered light from qubit A being absorbed by qubit B. Given the large separations of our qubits during measurement, such rescattered light is extremely weak compared to the measurement light itself, and so any correlations in the measurement errors are expected to be extremely small.

We also thank the reviewer for the suggested improvements, which we address below.

Suggested improvements

I cannot find many improvements from a scientific perspective for this work. I enumerate my minor concerns below.

On page 2 “A barrier is inserted before performing destructive qutrit measurements which ensures the entire quantum state is prepared...” Can the authors indicate more clearly that the barrier is a compiler command (if this is a correct statement).

Yes, the barrier is a command given to the compiler, which then makes sure to only send the instructions to measure after all state preparation steps have been completed. We have indicated this using: *“This compiler command ensures that the system reaches the full \mathbb{Z}_3 ground state wavefunction before measurements collapse it into a product state.”*

On page 3: I really like figure 2. I would suggest adjusting the second line on the torus image because the blue is slightly hard to read against the dark grey.

We have increased the brightness of the shading on the torus image. The logical expectation value is now more clearly readable.

On page 4: I would like a larger and broader discussion on the impacts for quantum computing in terms of systems that would benefit from qutrit based hardware, i.e. high energy physics, spin models etc.

We have added a few references to systems that would benefit from qutrit hardware in the Conclusion section by modifying the sentence *“Our findings show that digital quantum processors have advanced to the point where they can throw off the shackles of the underlying qubit architecture and explore the much larger class of systems that is naturally formulated with local Hilbert space dimensions greater than two, which includes lattice gauge theories [52, 53] and models with spin greater than one-half [54, 55].”*

Reviewer #2:

Summary of the key results:

The paper by Iqbal et al presents an emulation of a qutrit toric code based on using pairs of qubits to encode each qutrit. Using up to 56 qubits, they study the topological properties of the toric codes as well as defects within the code.

The manuscript demonstrates interesting physics and very nice experimental results in a regime that has so far been challenging to reach for quantum computers. This highlights the potential of near-term devices for the study of exotic phases of matter. However, the presentation is at times quite difficult to follow, since topics such as anyon braiding are very specialized. For a broad journal such as Nature Communications, a more pedagogical introduction is needed to fully grasp the results. More detailed comments below.

1) The title and abstract should be transparent about the fact that this is an emulation of a qutrit experiment, rather than a native qutrit experiment. With the recent push into qudit computing platforms, I believe this distinction is relevant so that the article will not be misunderstood. The actual hardware implementation is first mentioned only at the end of Sec 1, which is too late.

This is a good point and we do not want to confound the results in our work from platforms which natively operate on qutrit degrees of freedom. To reflect this, we have modified the sentence in the abstract: “Here, we prepare and measure a ground state of the Z_3 toric code state on 24 qutrits (obtained by encoding one qutrit into two qubits) in a trapped ion quantum processor with fidelity per qutrit exceeding 96.5(3)%”.

2) Line 59: “ $e \leftrightarrow m$ symmetry of the toric code” is undefined. What does e and m refer to here? Also, more explanation is needed regarding the anyon properties and operations. e.g. how is Eq. (5) and (8) to be understood?

We have added the definition of the $e \leftrightarrow m$ symmetry as well as what we mean by an e , m , or other anyon to be present: “...Majoranas, associated with the $e \leftrightarrow m$ duality symmetry of the toric code, which exchanges the role of flux and charge and which is realised by mapping the square lattice to its dual or, equivalently, applying a Hadamard gate on each qubit and translating by half a lattice spacing. Here, and throughout this work, we will refer to the presence of anyons (like e and m) whenever a stabiliser term in the Hamiltonian is violated, i.e. its expectation value is maximally different from the ground state value (+1)”.

We have also defined what we mean by the fusion equations Eq. (5) and (8): “By fusion outcome, we mean the resultant stabilizer measurement when two excitations are moved to the same location on the lattice.”

3) The color shading in the figures is chosen over a range from 0 to 1, while the values are generally between 0.9 and 1. Hence, the differences in shading are almost undetectable. I suggest shifting the range to make this more useful.

We agree that differences in shading are almost undetectable, except for Fig. 5 in the manuscript. However, setting the limits of the colour bar between 0 and 1 was a conscious decision that we are hesitant to change, for the following reason:

The 0 expectation value for a stabiliser carries a significant physical meaning, namely indicating that an excitation will reside there with certainty. The physical interpretation of the full stabiliser range is at the heart of why a number of previous papers in the field set the precedent to include the full range in the colour bar:

Science 374, 1237-1241 (2021)

Nature 618, 264–269 (2023)

Nature 626 505-511 (2024)

Nature Comm Physics 7, 205 (2024)

In fact, the relative uniformity of the colours is a communication device: A relatively low probability of spurious anyons everywhere in the ground state is a requirement to do more complicated experiments later on — checking the uniformity of the colour scale is an easy way for the reader to confirm that indeed the ground state preparation is relatively successful before moving on to the rest of the paper.

Finally, achieving such a uniform picture is far from trivial: Quantum devices even just a few years ago would have had a hard time producing anything uniform-looking even with respect to the full stabiliser range for the colour bars.

4) How are 56 ions enough to emulate 24 qutrits, which need 2 ions each? Could be a typo, or maybe I am missing something?

It is true that the trap actually holds $56 \cdot 2 = 112$ ions (56 Ytterbium, 56 Barium). The Barium ions are “cooling ions” that do not participate in the quantum information and only the Ytterbium ions are encoded into qutrits. This encoding uses 2 qubits per qutrit (using the encoding shown in

“Appendix A: Details about the Circuit Decompositions”), so the maximum number of qutrits would be $56/2 = 28 > 24$. The setup is described in detail in Phys. Rev. X 13, 041052 (2023).

5) Sec III / Fig. 3: More explanation is needed for the feed forward operation used in Fig.3a. More explanation is needed for the braiding experiment in Fig.3. More explanation is needed on how the application of a Z operator creates a charge anticharge pair. The change $e \rightarrow m$ mentioned in Eq.(6) appears to be simply due to the location of the charge: on green plaquettes it is e , on blue plaquettes it is m ? Under what circumstances should one expect a different outcome here?

We have added a sentence that defines the exact feed-forward operation we do to initialise the parafermion defect:

“We apply a feedforward operation based on the measurement outcome which ensures that the corresponding defect stabilizers have definite values. Specifically, when the measurement outcome is 1, we apply a $\mathcal{Z} \otimes \mathcal{Z} \otimes \mathcal{Z}^\dagger$ to the left, bottom-left and bottom-centre qutrit in \ref{fig:em_defect}. If the measurement outcome is 2, the conjugate is applied instead and no action in case of a 0 measurement outcome.”

What we are describing here can be visualised like this:

Regarding the change $e \rightarrow m$: This is exactly true: All green plaquettes are “X-type” (the associated stabiliser operator contains 4 X-operators and the blue plaquettes are “Z-type”. Excitations of the X-type are charges (e) and those of the Z-type are fluxes (m). In absence of the defect, it would be impossible to transmute these two different excitation types into one another, as they live in different superselection sectors. (This also occurs in the rotated toric code model (also known as the XZZX model) where each stabilizer is the same, but the difference between e and m anyons is determined by which sublattice is excited; e.g., see <https://errorcorrectionzoo.org/c/xzzx>)

We have added more explanation in the caption of Fig. 3: “A pair of charges, e and $ebar$, is created by applying a Z operator, which toggles the eigenvalues of the neighboring green (X)-type plaquettes. Charge e remains fixed, while e is dragged through the defect pair and

emerges as m on the other side of the defect pair, signaled by the fact a blue (Z-type) plaquette is now excited.

6) Sec IV: More explanation needed for the hatching in Fig. 1a. Why are these values not defined after the circuit? More explanation is needed for Fig.4h. In Fig 4j, it is mentioned that the pair is created at the top left corner, yet in the figure it appears to be one plaquette lower, which leads to confusion with the final fusion of the pair. In line 284 it is stated that the same outcome is obtained using "a single pair of CC defects." Please clarify what is referred to here. In the next line it is stated that Fig.13 "partially demonstrates the conjugation of ..." - why partially?

Regarding the hatching patterns throughout the manuscript: These refer to non-local operators, similar to logical operators in the toric code. Since they extend across the lattice, there is no straightforward way to visualise them on equal footing with the local stabilisers and its expectation values are unrelated to the presence of anyons. This is why, in the caption of Fig 4, we say: *"These stabilizers are marked by a hatching pattern with a 'x' symbol, and their values are omitted for clarity as they label the internal state of the defects which is not locally accessible."*

Regarding the dashed lines in Fig 4h: We have added a clarification in the caption of Fig 4h: *"The dashed line shows the path (as implemented by U) taken to fuse the defect pair"*.

Regarding the location of the anyon creation in Fig 4j, it is correct that the pair is actually created in the plaquette below the top-left corner. We have clarified this in the figure caption: *"The m flux from the pair m - m bar created in the second plaquette from the top, leftmost corner remains fixed, while its partner m bar anyon is commuted ..."*

Regarding the same outcome from a single pair of CC defects: What we are referring to here is the final outcome of the transformation $\bar{m} \rightarrow \bar{e} \rightarrow m$. We have clarified this by adding *"This is the exact same outcome we obtained for a similar anyon trajectory and a single pair of CC defects, which would similarly transform an $\bar{m} \rightarrow m$ when moved across"*.

Regarding the "partial demonstration of the conjugation": Defect lines are extended, line-like objects and action can be taken on any part of the line. This is what we chose to do in Supplementary Fig. 13, where *part of* an extended parafermion defect line is conjugated with a charge-conjugation defect line. We have clarified the sentence to read: *"In addition, Supplementary Fig. 13 demonstrates the conjugation of (part of) a parafermion defect line with a CC defect line, effectively resulting in a conjugate parafermion defect line "*

7) Sec V: The state description in line 305 is unclear. What does it mean that a "pair fuses to" e ?

Let us discuss the state $|e_1e_2\rangle$, for clarity: Each state hosts 2 charge-conjugation defect pairs at fixed locations on the lattice:

Now, readout of the logical information in a pair would require moving the end points (using “Appendix F: Coherently moving the end of a charge conjugation defect”) towards each other until they coincide. Now measure all the stabilisers. In the experimental protocol in question, there are only three options:

Option 1) All stabilisers attain their ground state expectation values. This is what we call $|1\rangle$.

Option 2) One (green) stabiliser has expectation value ω . This is what we call $|e\rangle$.

Option 3) One (green) stabiliser has expectation value $\bar{\omega}$. This is what we call $|\bar{e}\rangle$.

To clarify this definition, we have added a better description of what we mean by “fusion outcome” the first time the term appears: *“By fusion outcome, we mean the resultant stabilizer measurement when two excitations are moved to the same location on the lattice.”*

8) It would be nice to add some discussion about what one learns from these experiments.

The main thing that we wanted to learn in this work is this: Given that there are many applications that require qutrits (and in fact many developments in realising these qutrits natively), is it reasonable to try to simulate qutrit systems by encoding them into qubits? For the present application, the answer is “Yes, but the error is roughly 2-3 times of the qubit equivalent”: (Qubit toric codes can be prepared at least with fidelity per site $\sim 98.6\%$ vs the 96.5% reported here). While such error rates are still sufficient to see strong signals in qutrit simulations (the present work being an example of that, and we give examples of systems in the new Conclusion section below), an open question (which we hint at in the end) is whether this effective error rate increase allows one to do *fault-tolerant* qutrit quantum computation using elementary encoded qutrits. Resolving that question will require future work.

We were also surprised to learn that, in contrast with schemes based on measurement and feed-forward, the unitary state preparation scheme we used here shows a high degree of translation-symmetry breaking.

To make all of these learnings, we have revamped the Conclusion section which now reads like this:

“We have created high-fidelity ground states of the Z_3 toric code on tori of sizes up to 6×4 qutrits using Quantinuum’s H2 trapped ion quantum computer. We found that the unitary state preparation scheme employed in this work shows a much higher degree of translation-symmetry breaking than measurement-based preparation schemes, which could be used in heralded encoding protocols in future work. We have created parafermion and charge conjugation defect

pairs on top of this vacuum and verified their action on the anyons of the model; some of these operations were facilitated by using adaptive quantum circuits. Finally, we have initialised a topological qutrit from two charge conjugation defect pairs. An appealing direction for future work is to explore syndrome measurements and repeat-until-success protocols, which have thus far led to a break-even for state preparation and measurement errors for simpler code states [6, 7, 49–51].

Our findings show that digital quantum processors have advanced to the point where they can throw off the shackles of the underlying qubit architecture and explore the much larger class of systems that is naturally formulated with local Hilbert space dimensions greater than two, which includes lattice gauge theories [52, 53] and models with spin greater than one-half [54, 55]. While our results show that qutrit quantum simulations are now generally possible using qubit-based devices, the question of whether fault-tolerance thresholds for qutrit-based codes can be achieved in such devices remains an open question.

Tantalisingly, that class includes models with universal quantum computational power, some of which are closely related to the model presented here [25].”